# *Entia Non Sunt Multiplicanda* . . . Shall I look for clusters in my cognitive data?

**Enrico Toffalini**[1]*, **Paolo Girardi**[2,3], **David Giofrè**[4], **Gianmarco Altoè**[2]*

**1** Department of General Psychology, University of Padua, Padua, Italy, **2** Department of Developmental Psychology and Socialization, University of Padua, Padua, Italy, **3** Department of Statistical Sciences, University of Padua, Padua, Italy, **4** DISFOR, University of Genoa, Genoa, Italy

* enrico.toffalini@unipd.it (ET); gianmarco.altoe@unipd.it (GA)

**Data Availability Statement:** All data concerning Study 1 and the R code used for the simulation and its output concerning Study 2 are available on the Open Science Framework data base (https://osf.io/quncr).

## Abstract

Unsupervised clustering methods are increasingly being applied in psychology. Researchers may use such methods on multivariate data to reveal previously undetected sub-populations of individuals within a larger population. Realistic research scenarios in the cognitive science may not be ideally suited for a successful use of these methods, however, as they are characterized by modest effect sizes, limited sample sizes, and non-orthogonal indicators. This combination of characteristics even presents a high risk of detecting non-existing clusters. A systematic review showed that, among 191 studies published in 2016–2020 that used different clustering methods to classify human participants, the median sample size was only 322, and a median of 3 latent classes/clusters were detected. None of them concluded in favor of a one-cluster solution, potentially giving rise to an extreme publication bias. Dimensionality reduction techniques are almost never used before clustering. In a subsequent simulation study, we examined the performance of popular clustering techniques, including Gaussian mixture model, a partitioning, and a hierarchical agglomerative algorithm. We focused on their ability to detect the correct number of clusters, and on their classification accuracy. Under a reasoned set of scenarios that we considered plausible for the cognitive research, none of the methods adequately discriminates between one vs two true clusters. In addition, non-orthogonal indicators lead to a high risk of incorrectly detecting multiple clusters where none existed, even in the presence of only modest correlation (a frequent case in psychology). In conclusion, it is hard for researchers to be in a condition to achieve a valid unsupervised clustering for inferential purposes with a view to classifying individuals.

*Entia non sunt multiplicanda praeter necessitate*

*[Entities should not be multiplied beyond necessity]* (William of Ockham's razor)

## Introduction

Ockham's razor introduced a principle of parsimony in scientific reasoning: we should not engage in describing multiplicities unless it is clearly necessary. In the cognitive research, and

**Funding:** This study was supported by a grant from the MIUR (Dipartimenti di Eccellenza, DM 11/05/2017 n.262) awarded to the Department of General Psychology, University of Padua. The funder had no role in study design, data collection and analysis, decision to publish, or preparation of the manuscript.

**Competing interests:** The authors have declared that no competing interests exist.

in psychology or in the social sciences in general, a fundamental issue concerning multiplicities lies in whether individuals can be grouped around discrete taxonomies rather than just being seamlessly distributed along continuous dimensions. The issue can become delicate when clinical implications are at stake. In cognitive psychology, for instance, even clinical diagnoses based on widely accepted criteria (like those of the DSM-5) [1] could reflect a sensible, but somewhat arbitrary categorization of individuals based on cut-offs along dimensions that are naturally continuous. An example concerns the recently (re)opened debate on the multiplicity of learning disorders [2–4]. According to the dimensional hypothesis, individuals with most neurodevelopmental disorders may be more likely to represent the ends of developmental (multivariate) continua that encompass the general population, rather than internally homogeneous clusters of cases that are distinguishable from one another. Unfortunately, without clear and near-deterministic psychological, genetic, or neurological markers, it is hard to say whether some diagnoses reflect a natural taxonomy rather than a categorization that is convenient for practical purposes.

To disentangle this issue of multiplicity, one promising approach is to look for discontinuities in the data. For instance, if scores on certain indices are distributed over a continuum, and tend to consistently group around certain centroids, leaving a limited number of statistical units in the multidimensional "no man's land", this distribution may reveal hitherto-undetected underlying clusters in a non-homogeneous population. The procedure described relies on an unsupervised machine-learning algorithm clustering that—in the psychological and social sciences—is part of the methods used in latent profile analysis (LPA) [5]. A model-based clustering is often adopted in LPA, generating results that are particularly engaging because researcher can obtain a marginal and a joint probability distribution of the indicators for each latent class, characterizing the discovered sub-population [6, 7]. In addition, model-based clustering also has the potential to deal with the issue of causality [8]. In LPA, latent classes can be interpreted as the underlying causes of mixture distributions observed in the indicators, in a similar way as latent variables can be interpreted as being causally related to observed indicators in measurement models [e.g., 9]. For the sake of clarity, an (oversimplified) example is given by the political polarization in the United States of America (USA). Republican and Democratic voters have tended to voice, on average, different opinions about adherence to measures against COVID-19 and vaccinations [10–12]. If political convictions are among the causes behind their different beliefs/attitudes, then mixture distributions should be observable in the latter, revealing two underlying clusters. An outside observer who knows nothing about the political polarization in the US might note this and they would correctly infer the existence of two clusters from the observed indicators alone. Alternatively, if the direction of causality were different from the one mentioned above, then clusters might not emerge. For example, taking a political stand could be a consequence (rather than a cause) of one's beliefs/attitudes regarding a series of issues including those listed above. In the latter case, mixture distributions may not be observed in beliefs/attitudes, and the latter may better be described as a multivariate continuum.

Clustering techniques and LPA are gaining popularity in psychology. Searching for "clustering" or "cluster analysis" or "latent profile analysis" in titles, abstracts or keywords, in areas of the Social Sciences and/or Psychology, in the Scopus database revealed 19,541 records in the years 2010–2020, as opposed to just 9,404 in all the years prior to 2010 (the ratio is 2.08). The total number of records in Social Sciences and/or Psychology was 3,824,683 in 2010–2020 and 3,570,607 in all years prior to 2010 (the ratio is 1.07). Therefore, publications mentioning clustering/LPA have grown at nearly twice the rate of the total publication count in the last 10 years as compared to all previous history. This may also reflect how easy it has become to apply clustering techniques using the various software options available.

Unfortunately, data from research in cognitive psychology may not be well suited to clustering techniques. A first major issue is the plausible effect size, which tends to be rather small in psychological studies. Large replication studies have suggested that the true average effect size may equate to a correlation *r* of .20-.25 [13–15], which is equivalent to a standardized mean difference between groups of about 0.40–0.50. Modest effect sizes probably reflect the large inter- and intra-individual variability in psychological phenomena (measurement error of variables may also further attenuate the effect sizes). In psychology, large effect sizes (e.g., Cohen's d >0.80) may be due to an inflated estimation deriving from an underpowered design [e.g., 16], possible combined with a publication bias. For example, Schäfer and Schwarz [17] showed that while the median effect size in randomly selected publications without pre-registration may be large (*r* = 0.36, equivalent to *d* = 0.77), publication with pre-registration report substantially lower median effect sizes (*r* = 0.16, equivalent to *d* = 0.32). Alternatively, large effect sizes may reflect an already well-known, obvious association between variables (e.g., the relation between two areas of intelligence). According to Cohen [18], *d* = 0.50 is an effect "likely to be visible to the naked eye of a careful observer". We speculate that a *d* > 1.00 would be visible even to the non-expert eye, with statistical inference being used simply to endorse the obvious. To give an instance of a very large effect size, Cohen's *d* for the difference in height between young adult males and females is nearly 2.00, based on WHO data [19]. This is hardly the magnitude of effect size that we can plausibly expect from genuine discoveries in psychological research in cognition, however. Standardized mean differences of such magnitude could be found, for example, in clinical *versus* control populations when they are compared on criterion variables that led to a diagnosis, or on strictly related scores. Examples could be reading scores for children with dyslexia versus controls, or IQ scores for children with an intellectual disability versus those with a typical development, where *z* < -2 is generally used as the clinical cut-off. Such comparisons would clearly be meaningless, however. In conclusion, we suggest that a standardized mean difference of around 0.40 could be regarded as a plausible effect size when comparing different groups on behavioral measures in cognitive psychology and related fields (this is just an educated guess based on the general literature, however, and it may largely vary in specific research sub-fields.)

On the topic of power analysis in clustering techniques, the minimum effect size for algorithms to detect the number of clusters correctly would be considered quite large in cognitive research. Tein et al. [20] concluded that at least *d* = 0.80, with latent classes differing in at least 10 orthogonal indicators, and a large sample is still needed (N ≥ 500), is required for LPA. Recently, in a large simulation study, Dalmaijer et al. [21] showed that clustering techniques may work with as few as 20–30 observations per cluster. However, this only happens when clusters present standardized Euclidean distances between cluster centroids (after projection into two dimensions) of at least Δ = 3–4. This requires, for example, *d* = 0.80 in 15–25 orthogonal indicators, or *d* = 1.50 in 4 orthogonal indicators. These effect sizes should be considered as large to extremely large in the area of cognitive research.

Another problem concerns dimensionality. We can assume that a researcher planning new research in cognitive psychology would carefully select relatively limited numbers of measures to administer to avoid unnecessary redundancies and ensure feasibility of data collection. Even so, correlations between continuous indicators could still be a problem because clustering techniques like LPA assume that the indicators are orthogonal [e.g., 20]. Assumptions of orthogonality may be an issue in cognitive psychology. To give an example, all measures involving cognitive performance tend to correlate positively, a phenomenon known as the "positive manifold" in the field of intelligence (which is generally thought of as reflecting a common "*g* factor" underlying all measures) [22, 23]. However, it may be difficult to establish *a priori* precisely how much correlation one should expect, unless a standardized

comprehensive battery with robust normative data is used. Other dimensions reflecting personality or clinically relevant factors may also correlate naturally in the population to degrees that are unknown *a priori*. This means that, when obtaining clusters from multivariate data, we may have trouble establishing, for example, whether an emerging correlation between a few variables reflects underlying clusters that differ on two orthogonal indicators or just two variables continuously correlated in a given population. Techniques for dimensionality reduction (e.g., exploratory factor analysis, principal component analysis [PCA]) could be the key to reduce a large set of correlated variables to a smaller set of orthogonal dimensions before clustering (see also Dalmaijer et al., [21], for important insights on this point). However, it is unclear how frequently these are effectively used in the extant literature.

It is important to ascertain the conditions under which clustering methods can detect true underlying clusters or latent classes because, under suboptimal scenarios, we risk not only failing to identify the true clusters, but also to infer the existence of sub-populations where none exist. To disclose some results in advance, correlated clustering variables easily lead to inflating the number of clusters detected. For example, simulating data with only one true latent class, 8 clustering variables correlated at $r = 0.30$, sample size of N = 200, and performing model-based Gaussian mixture clustering with the Bayesian Information Criterion (BIC) [24] for model selection, led to identifying 2 to 4 clusters as optimal solutions 100% of times over 1,000 iterations (see Study 2 for details on the simulation procedure). In such cases, the identified clusters generally differ only in terms of their overall mean score in most or all clustering variables simultaneously (e.g., overall low vs middle vs high). A typical example is illustrated in Fig 1. A similar observation, concerning latent growth classes estimated from growth mixture models simply segmenting the true continuum of individual differences into pieces, was made by Bauer [25]. Negatively correlated variables simply lead to reversals in the mean scores.

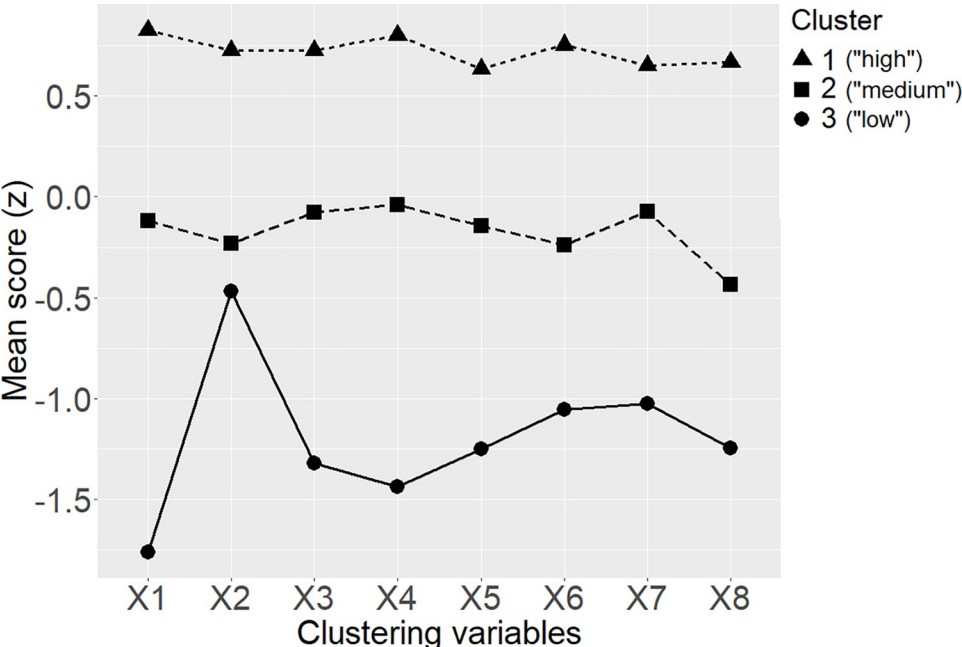

**Fig 1. Example of the typical profiles of mean scores of clusters detected from a set of positively correlated variables.** Clustering variables were correlated with r = 0.30, normally distributed, and without any true underlying sub-populations. Data was simulated with N = 200. Model-based Gaussian mixture clustering was performed, with BIC used for selecting optimal number of clusters (see Study 2 for details).

In the present study, we examined the conditions under which popular techniques for clustering individuals may plausibly work on cognitive data. We systematically examined the role of effect sizes, sample sizes, correlations between indicators, and number of indicators, on the performance of clustering techniques. As regards the inferential risks, we focused not only on power (i.e., the probability of detecting the true number of underlying clusters under a given scenario), but also on false positives (specifically, the risk of detecting multiple clusters where only one underlying population exists). In other words, we examined the case in which clustering techniques may mislead researchers, inducing them to conclude that there are multiple sub-populations that do not exist. We also conducted a preliminary review of the literature to illustrate the most common characteristics and practices of recently published studies that used clustering techniques on cognitive/behavioral data to isolate subgroups of individuals from larger samples.

As in previous analyses, we examined the performance of clustering techniques on simulated data, defining an extended set of scenarios. The novelty of the present study lies in that, in addition to analyzing the effect of factors such as sample sizes, effect sizes and number of variables, we also tested different sizes of correlations among indicators. We opted to concentrate on a set of well-reasoned, plausible scenarios in cognitive research, and on our interpretation of them. Our findings may be useful in any other area of psychology where domains and related indicators are often correlated. In fact, the simulation examined in the present work considered sample sizes, effect sizes, numbers of indicators, and correlations that are appropriate for reflecting the most typical characteristics of empirical research. The parameters considered were partly drawn from the results of the preliminary literature review (Study 1).

We adopted three popular unsupervised clustering techniques–the Gaussian mixture model (Fraley at al., 2012), a partitioning algorithm (Partitioning Around Medoids [PAM]; Kaufman and Rousseeuw [26], and a hierarchical agglomerative clustering (HAC) algorithm–that cover the algorithms most often used in real observational studies (Study 2). The k-means algorithm was not reported because its results were virtually redundant with those of PAM. The model-based procedure assumed that the indicators are normally distributed, taking into account a correlation structure between indices. The optimal number of clusters is commonly defined on the basis of a trade-off between the variance explained and the model's complexity, like the BIC, for instance. The PAM and HAC algorithms do not assume any particular distribution for the indices, and clustering is commonly based on a dissimilarity matrix, i.e., the Euclidean distance calculated between the raw data. The PAM algorithm chooses a set of statistical units, called medoids, and iteratively searches for an optimal allocation of the other units around these medoids. The HAC algorithm aggregates the most similar statistical units, taking an agglomerative iterative procedure. Using these last two procedures, the number of clusters is commonly defined on the basis of a measure of intra-cluster entropy, or the ratio of the variability of the indices between and among the clusters [27, 28]. For a quick example on the features of the three clustering methods, see the section "Clustering procedures" below.

## Study 1: Literature review

We aimed to investigate the features and common practices shared by published studies using clustering techniques applied to behavioral data. Our goal was to obtain a comprehensive sample of publications in which clustering methods (including LPA) were employed, focusing on the psychological literature involving cognitive and neuropsychological methods and measures. This approach was taken because we were particularly interested in the use of clustering methods in studies within a cognitive framework.

## Method

**Literature search and inclusion criteria.** The Search function in Scopus was used, with the following keywords: ("clustering" or "latent profile analysis" or "LPA") and ("cognition" or "cognitive" or "neuropsychology" or "neuropsychological") in titles, abstracts or keywords. The results were further limited to three subject areas, "Psychology", "Social sciences", and "Neuroscience", and to studies published over the 5 years between 2016 and 2020 (included).

Rather liberal inclusion criteria were adopted. We included any study performing clustering analyses (including LPA) on behavioral data within a cognitive psychology framework (e.g., scores from performance test, self-reported measures). Studies performing clustering on physiological measures or neuroimaging data only were excluded (as they have features that go beyond the scope of the present report), and so were papers involving non-human subjects. Only studies in which the optimal number of clusters was (or appeared to be) determined based on the data via the clustering algorithm were considered.

**Coding of the studies.** The following information was encoded for each study: title, authors, and year of publication; sample size on which clustering was performed; number of indicators; clustering algorithm used, and criteria adopted to identify the number of clusters or latent classes; the number of alternative solutions considered; the number of clusters or latent classes detected; and whether the profiles of mean scores differed between clusters mostly in terms of mean levels across most variables simultaneously (based on the inspection of figures if reported, or on the very interpretation of the authors of the study). We considered the following characteristics as indicators of good/questionable practices in the specific context of cluster analysis: whether the one-cluster solution was tested; whether the number of alternative solutions considered was explicitly stated; whether any technique for dimensionality reduction was preliminarily used (on the whole set of variables/indexes, and before performing clustering); whether the cluster analysis was preregistered; and whether power for the cluster analysis was declared *a priori*.

If the same study performed more than one clustering or LPA on the same or a different sample reaching different conclusions, only the first one was reported, for simplicity and to avoid dependencies in the coded data. Only papers in English, Italian, Portuguese, Spanish or French were considered. Only studies in which the final number of clusters was inferred from the data were included for further analysis.

Additionally, we examined the characteristics of the journals in which the papers were published. We looked at the Clarivate Journal Citation Reports (JCR; https://jcr.clarivate.com/jcr/home). We considered their Impact Factor (IF) in 2020 (the latest available as of February 2022), and their category/ies and relative quartile/s as a proxy of the journal's quality.

## Results and discussion

The PRISMA flow diagram of the study selection process is contained in the Supplemental online material, S1 Fig. There were ultimately 191 studies included in the quantitative synthesis.

Preliminarily, we report the characteristics of the journals in which the studies were published. As of February 2022, 179 (94%) was published in a journal indexed by the JCR with an IF. The median IF was 3.23. Overall, high quartiles were overrepresented: Q1 was represented 41% of times; Q2, 32%; Q3, 23%; Q4, 4%. The five most represented categories were: Psychiatry (29% of indexed studies); Psychology, Developmental (20%); Psychology, Clinical (13%); Psychology, Educational (11%); Clinical Neurology (11%). Concerning the characteristics of the clustering conducted in the studies, Table 1 shows the percentiles of interest for sample size (i.e., number of participants on which clustering was performed), the number of variables

**Table 1. Percentiles of interest for the number of individuals on which clustering was performed, the number of indicators, and the number of clusters identified, across the 191 studies reviewed.**

| | Percentiles | | | | |
|---|---|---|---|---|---|
| | 5th | 25th | 50th | 75th | 95th |
| N. of individuals on which clustering was performed | 66 | 153 | 322 | 589 | 2,119 |
| N. of variables (indicators) used for clustering | 3 | 4 | 6 | 9 | 19 |
| N. of clusters identified | 2 | 3 | 3 | 4 | 6 |

used for clustering, and the number of clusters or latent classes detected, across the 191 studies reviewed. The "median" study performed cluster analysis on 322 individuals, used 6 indicators, and detected 3 clusters, while most studies (between the 5th and 95th percentiles) performed cluster analysis on numbers of individuals ranging from 66 to 2,119, used between 3 and 19 indicators, and detected from 2 to 6 clusters.

Model-based clustering algorithms (including LPA and LCA) were the most often used (146 studies; 76%), which however may also reflect our search keywords. Next, hierarchical clustering was used in 21 studies (11%). A partitioning clustering method (specifically, k-means) was used in 18 studies (9%). The remaining cases used other methods.

As concerns the good/questionable practices concerning clustering, only 86 studies (45%) explicitly stated that they tested the one-cluster solution; 65 (34%) overlooked this solution and started testing from two clusters; while the remaining 40 studies (21%) did not explicitly report information on the number of possible solutions tested (i.e., the number of alternative clusters or latent classes examined). When reported, the number of possible solutions tested ranged between 1 and 15 clusters, with one exception testing a 36-cluster solution. Dimensionality reduction techniques were rarely used: 184 studies (96%) used none or at most locally (i.e., only for specific domains, but not for the whole set of variables used for clustering). Two studies used PCA, and other 4 used either exploratory or confirmatory factor analysis (note that the latter do not ensure extracting uncorrelated factor scores, however). Finally, only three studies were preregistered, and only four studies explicitly included a power analysis (which confirms that the inclusion of an a priori power analysis is often neglected [29]), but always for more general analyses. That is, none of the 191 studies performed a power analysis specifically for clustering. This confirms the observation that papers in psychology, even if published in high ranking journals, do not necessarily conform to high statistical standards [30] and are rarely pre-registered [31]

Finally, 92 studies (48%) presented clusters whose average scores were clearly dominated by an overall-low vs an overall-high profile (with other possible profiles in the middle). Among studies detecting only 2 or 3 clusters, this occurred even more frequently (60 out of 102 studies, i.e., 59%). As explained in the Introduction, the latter is a typical warning flag that multiple clusters may have been detected where none exist, due to correlated clustering variables (Fig 1).

## Study 2: Performance of clustering algorithms under different scenarios

### Simulation settings

**Strategy.** The data were simulated using a multivariate Gaussian random variable distributed as follows:

$$Y|G = g \sim N_p(\mu_g, \Sigma(\rho_g))$$

where G is the true latent group or class, $p$ is the dimensionality, which equates to the number of indicators, $\mu_g$ is the vector of the mean for the cluster $g$, and $\Sigma(\rho_g)$ is the variance-covariance of the cluster $g$, a unit diagonal matrix with a pairwise correlation between indicators equating to $\rho_g$. In our setting, we fixed $\rho_g = \rho$ in all groups. Given the results of the preliminary literature review, we tested $p = 3$, 6, and 12 indicators, while the number of true clusters was either 1 (i.e., only one population, containing no sub-populations) or 2. This enabled us to simplify the analysis and related discussion, while more complex scenarios could easily be built on it *ad hoc*. The joint distribution of the indicators thus becomes:

$$Y \sim N_p(0, \Sigma(\rho)), \text{ with 1 group}$$

$$Y|G = g \sim N_p(d\, I_2(g), \Sigma(\rho)), \text{ with 2 groups}$$

where, in the simulation with two clusters, $I_2(g)$ is an indicator function taking a value of 1 when g = 2, and 0 otherwise. In the latter scenario, we explored three increasing levels of the standardized mean difference, with Cohen's $d = 0.40$, 0.80, and 1.50, which respectively correspond to a plausible average (small) effect, a large effect, and an overly large (implausible) effect size in the cognitive research. It is important to point out that a condition of d = 1.50 was considered solely for the purpose of examining whether the algorithms would work with exaggeratedly optimistic expectations.

Three different average values were chosen for the pairwise correlation ($\rho$): i) zero, to simulate the presence of orthogonal indicators (i.e., the assumption that, after controlling for cluster membership, the clustering variables are uncorrelated); ii) 0.20, which is a plausible modest correlation between any pair of variables (that may also reflect a partial correlation remaining between a pair of broadly overlapping variables even after controlling for a higher-order factor, e.g. between two cognitive variables after controlling for a measure of the $g$ factor); and iii) 0.50, which is a strong correlation (e.g., the correlation that could be expected between two uncorrected measures of cognitive performance).

For greater realism, and to reflect the plausible uncertainty that we might have when performing a real *a priori* design analysis, we sampled the parameters from probability distributions rather than having them as fixed values. For each clustering iteration, we therefore sampled Cohen's $d$ and the value of the correlation parameter $\rho$ from a uniform distribution taking the following form:

$$d \sim U(a, b) \text{ and } \rho \sim U(c, d),$$

where the pairs of parameters (a, b) and (c, d) were chosen to center the uniform distribution on the previously defined values. In detail, (a, b) were (0.20, 0.60), (0.60, 1.00) and (1.00, 2.00), while (c, d) were (-0.10, 0.10), (0.10, 0.30) and (0.40, 0.60). A condition in which the true bivariate correlation between two variables was exactly zero was never considered because we posit that such a situation can virtually never be assumed in cognitive psychology. Instead, we adopted a condition in which the absolute value of the correlation is below .10, which can be considered a range of negligible effects.

For each combination we tested six different sample sizes (N = 50, 100, 250, 500, 1000, and 2000), covering a wide range of cases observed in psychological research. In the case of the simulation with two true latent classes, two equally numerous sub-populations were simulated within the general population, with each statistical unit having the same probability of being sampled (thus, on average, $n_1 = n_2 = N/2$). The combinations of the number of indicators, the parameters $d$ and $\rho$, and the sample size gave rise to a total of 54 and 162 combinations for the scenario with one and two clusters, respectively.

**Clustering procedures.**    Three standard, different and competitive classification methods were considered: a) a model-based Gaussian mixture clustering (MGC); b) a Partitioning Algorithm around Medoids (PAM; in the context of our simulation, PAM yielded very similar results, but more robust than k-means clustering); and c) a Hierarchical Agglomerative Clustering (HAC) with a procedure based on a complete linkage algorithm that had achieved a good performance [32].

The main characteristics of the three clustering methods are here presented through a quick example of LPA to identify groups considering questionnaire data from the 2015 Program for International Student Assessment (PISA; https://www.oecd.org/pisa/data/) in the United States that can be retrieved by the R package *tidyLPA*. Each student was investigated through a questionnaire producing four self-reported composite quantitative variables (indexes): broad interest, enjoyment, instrumental motivation, and self-efficacy. We considered the responses of a random sample of 200 students out of 5,712 participants. The four indexes can obviously be expected to positively correlate to some extent even in a homogeneous population (i.e., one without any discrete cluster in it). Indeed, they show a medium-to-strong correlation structure with a statistically significant pairwise correlation among all the variables and a maximum correlation between Broad interest and Enjoyment (S2 Fig; Pearson's correlation: 0.523, which is strong).

The MGC method considers each characteristic as a finite mixture of normal random variables, each one belonging to a latent group. Varying the number of groups, the procedure selected the best type of gaussian mixture which meets a computational criterion (i.e., BIC). In our example, the optimal number of latent groups was 4, with a variance-covariance matrix structure defined as VVE (ellipsoidal, equal orientation; S3 Fig, left). The results of the clustering are presented in S3 Fig (right) and denoted a separation among the groups, especially for the values related to the Enjoyment, while the separation for the other variables is less visible.

The second method is based on the PAM algorithm. It starts from the calculation of a distance matrix between the statistical units, based on a metric norm; the L2-norm (the Euclidean distance) is commonly used. Consequently, fixing in advance the number of latent groups, the algorithm starts to search for a pre-determined number of representative observations, called medoids, among the observations of the data set. For each medoid and at each step, the algorithm selects the nearest observation and evaluates the reallocation of the previously identified medoid with the non-medoid observation. The algorithm ends when all the observation are allocated to a group. Different criteria have been proposed to choose the optimal number of groups. In our example we considered the commonly used silhouette value, which is a measure of cohesion and indicates how similar an observation is to its own cluster vis-à-vis other clusters. The silhouette value ranges from −1 to +1, where a high value indicates that the object is well matched to its own cluster and poorly matched to neighboring clusters. The best partition maximizes the average silhouette width. In our example, 2 groups/clusters provided the highest value for this index (S4 Fig).

The last clustering method is based on a Hierarchical Agglomerative Clustering (HAC) which is an iterative classification method which starts considering all the observations as a single-town cluster. As for the PAM approach, based on a distance matrix, the two observations whose clustering together minimizes an appropriate agglomeration criterion are then clustered together. The most frequently used agglomeration criterion are the complete distance, the average distance, and Ward's distance. Like the PAM algorithm, the number of clusters is selected maximizing the average silhouette value. Considering the PISA dataset, we reported a dendrogram based on the complete linkage which identified 2 clusters as the optimal solution (S5 Fig). In conclusion, in this example the three methods suggested a different number of groups/clusters as optimal (4 for MGC and 2 for both PAM and HAC), as well as dissimilar classification results.

In our simulation procedure, we assessed the performance of each clustering method with 500 iterations in each scenario, testing a number of clusters which varied from 1 to 5 at each iteration. The optimal number of clusters was chosen based on the BIC value for the MGC (BIC is a best-performing criterion according to Tein [20]) Tein, automatically testing different types of variance-covariance matrix, while for the HAC and PAM the number was selected by maximizing the average silhouette profile index [33]. Given that the average silhouette index cannot be calculated when the number of clusters equals 1, for the HAC and PAM methods the possibility of a single cluster was tested with the Duda-Hart test [34], taking a critical $\alpha$ = .05 for statistical significance (i.e., the significance threshold most widely used in psychological research), and comparing the solution formed by two clusters with the one formed by no clusters. The clustering methods were replicated 500 times for each previously-defined combination. For each iteration, we tested whether the algorithm detects the right number of clusters, and the proportion of statistical units correctly classified, using the Rand Index [35]. Correct classification performance was assessed from the proportion of right numbers of clusters, and the average Rand Index and its standard deviation (SD). The analysis was performed with the R free software (version 4.0) [36] and the packages: *mclust* [37] for MGC; *fpc* [38] for the PAM algorithm; *cluster* [39] for the HAC procedure; and *ggplot2* [40] for creating all figures.

**Computational costs.**   Regarding the computational cost, our simulation took 36 and 112 hours for the one- and two-cluster simulations, respectively, using a laptop equipped with a quad-core Intel i5 processor at 1.6 GHz and 8 GB RAM. The computational time could be significantly shortened via parallelization, but we abandoned this option after a preliminary attempt because it led to random number generation failing when a Windows operating system was used. The worst-case time complexity of the HAC method with complete linkage is O ($n^2 \log[n]$), while for the PAM it varied from O($n^2$) to O(n log[n]). The MGC depends on the convergence of the EM algorithm, which commonly takes a time following O(G p N), where G is the number of clusters considered, and p is the number of indicators. The EM algorithm in the *mclust* R library tests, by default, a wide range of possible variance-covariance matrices, choosing the one that minimizes a defined criterion (i.e., BIC index); this process implies an increased computational time.

## Results and discussion

Fig 2 shows the probabilities of inferring the existence of one (correct) latent group, two groups, three groups, and more than three. Such probabilities were very high and almost identical across the three clustering methods for all the sample sizes considered, but only in the presence of nearly orthogonal indicators. In this latter condition, the only exception was represented by the MGC, which tended to erroneously identify two groups with a moderate frequency, when the sample sizes are very large (N > 1,000). When we considered the condition in which there was a weak correlation ($\rho \approx .20$), the performance of all three methods clearly deteriorated, especially in the case of a large number of indicators and a large sample size. The best results were obtained by the MGC and the HAC algorithm for three indicators, resulting in a probability to detect one cluster and an average Rand Index greater than 60% and 80%, respectively. In the presence of a strong correlation ($\rho \approx .50$), only the MGC presented a high probability to detect one cluster and an elevated Rand Index, but only for at least 250 statistical units.

In the presence of two latent groups with a standardized difference, Cohen's $d \approx 0.40$ (Fig 3), the PAM and HAC methods had an average Rand Index showing that the classification did not differ from chance even in the scenarios in which they (apparently) present high power in

## One true cluster

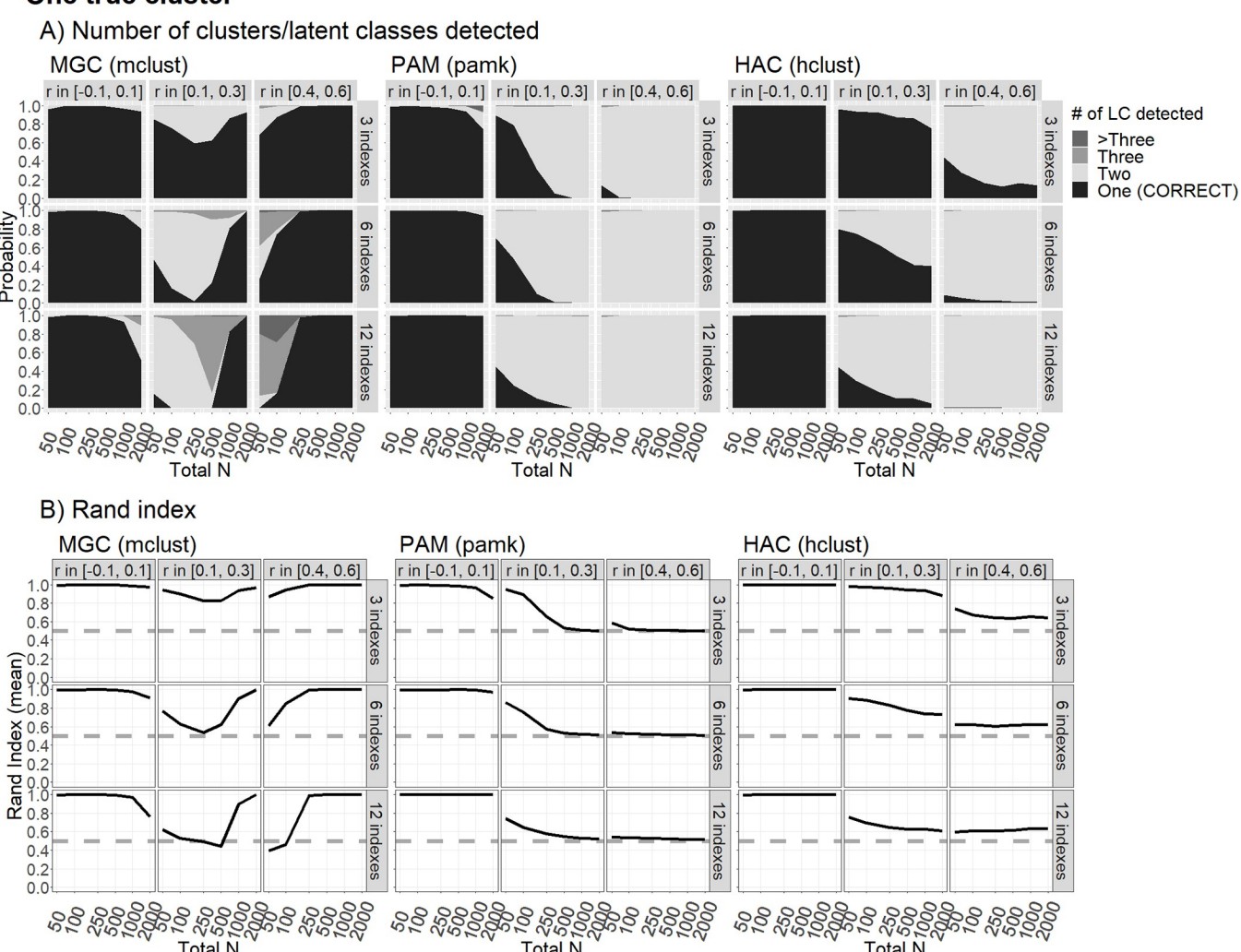

**Fig 2. Clustering performance with only one true cluster/latent class.** The figure shows clustering performance in correctly classifying one true cluster/latent class using the Model-based Gaussian mixture Clustering (MGC), Partitioning Around Medoids (PAM) and Hierarchical Agglomerative Clustering (HAC) methods, in terms of the probability to detect that there is one latent class, and the average Rand Index, as a function of sample size (N) and correlation I(r) among indices, over 500 replications. Words in parentheses indicate the R functions used.

detecting the right number of classes. This is due to such high power being driven by the presence of correlated indicators, which leads both methods to identify the wrong clusters. The MGC generated only slightly better classification results, but only in the case of very large samples and a large number of indicators. Performance improved slightly in the presence of a large effect size, d ≈ 0.80 (see Fig 4), giving rise to a high power in identifying the right number of groups, but only for the MGC and the PAM algorithm, and only for samples of at least 250 subjects. The Rand Index only reached values higher than 80% for the MGC based on 12 indicators, and in the presence of either no correlation ($\rho \approx 0$) or a high correlation ($\rho \approx .50$). In this last condition, with an average Cohen's *d* of about 1.5 (see Fig 5), all the methods performed well in terms of power and the probability of correct classification except in the cases of: the MGC with a low correlation ($\rho \approx .20$) and 12 indicators; and the HAC with a high correlation ($\rho \approx .50$).

The full set of results is available in the S1–S4 Tables.

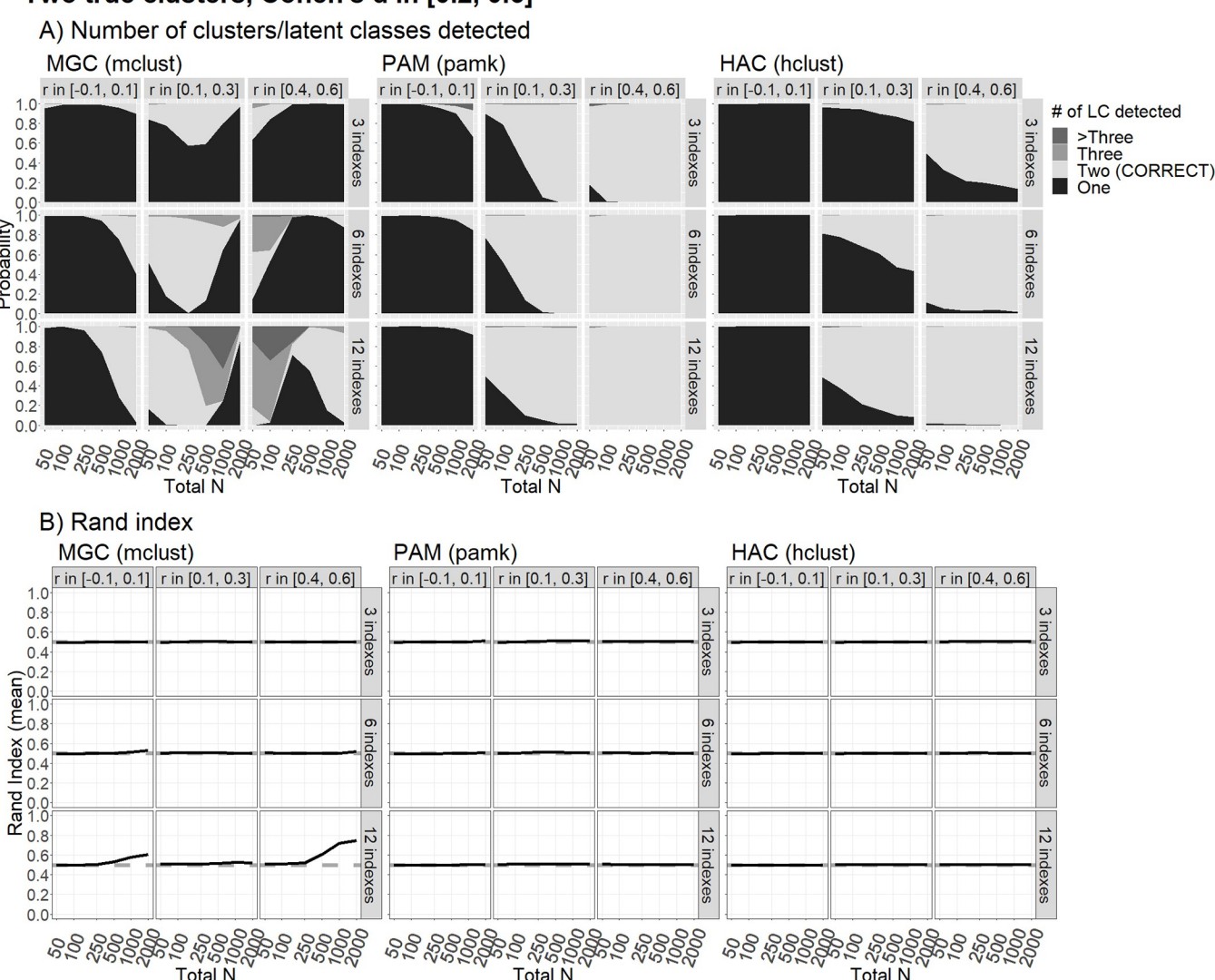

**Fig 3. Clustering performance with two true clusters/latent classes and small differences.** The figure shows the clustering performance in correctly classifying two true clusters/latent classes differing by a small-to-medium Cohen's d [0.2, 0.6] across the indices, using the MGC, PAM and HAC methods, in terms of the probability to detect that there are two latent classes, and the average Rand Index, as a function of sample size (N) and correlatiIon(r) among indices, over 500 replications. Words in parentheses indicate the R functions used.

Back to the example based on the PISA data with N = 200 (see the Clustering Procedure section), we may suspect that all solutions inflated the number of clusters, due to the 4 indexes being correlated. MGC detected 4 clusters, while PAM and HAC detected 2 clusters. Those were real data, so we cannot know the actual number of underlying clusters. However, Fig 2 shows that considering the set of scenarios with modest (r in [0.1, 0.3]) or strong (r in [0.4, 0.6]) correlations, with between 3 and 6 indexes, and N between 100 and 250, MGC has a high risk of detecting 2, 3, or even than 3 clusters, and PAM and HAC have a high risk of detecting exactly 2 clusters, even when there is only one true cluster in the population.

## General discussion

This study aimed to review the characteristics of a representative sample of published studies that had used clustering techniques on behavioral data. Based on them, we assessed the

## Two true clusters, Cohen's d in [0.6, 1.0]

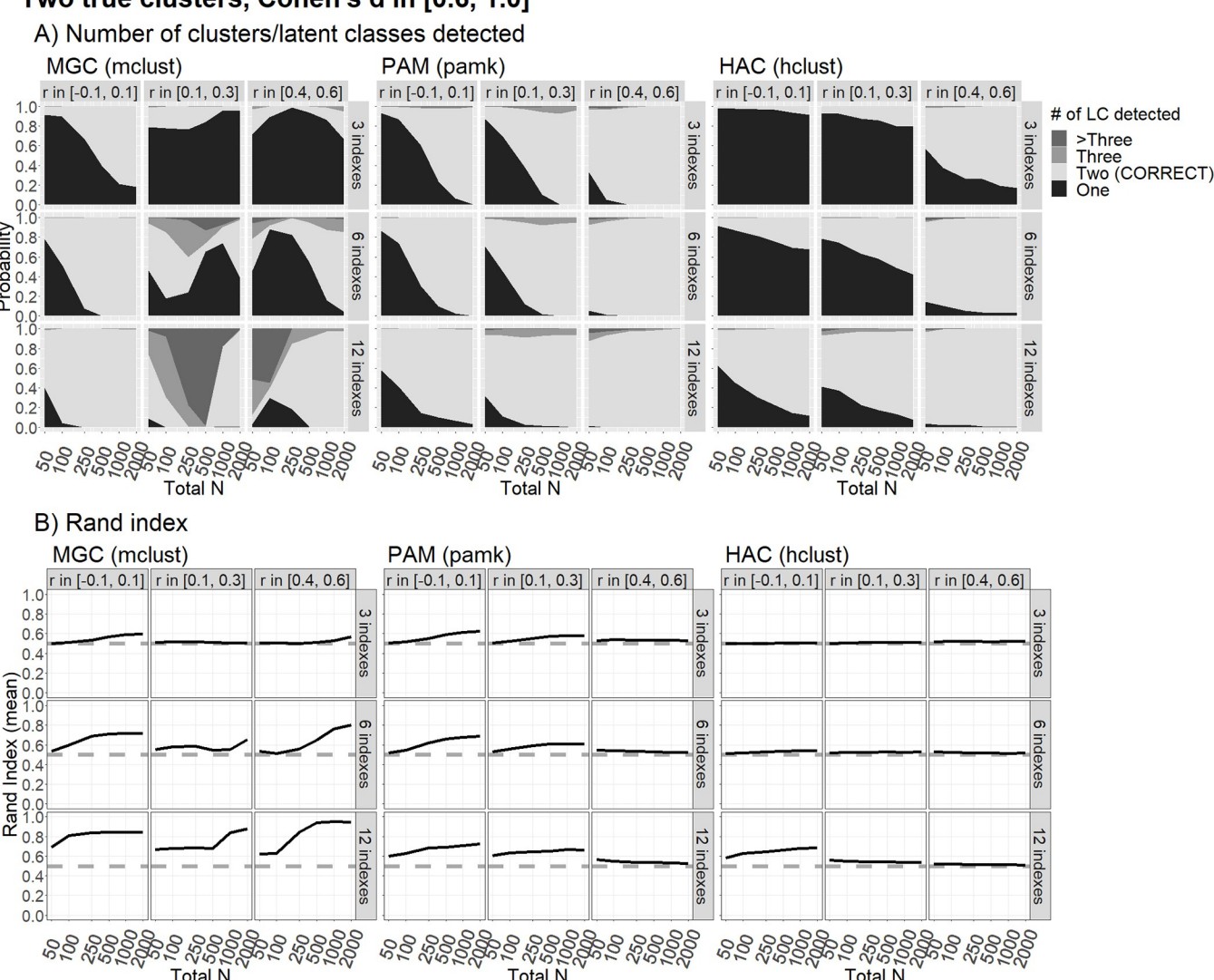

**Fig 4. Clustering performance with two true clusters/latent classes and large differences.** The figure shows the clustering performance in correctly classifying two true clusters/latent classes differing by a large Cohen's d [0.6, 1.0] across the indices, using the MGC, PAM and HAC methods, in terms of the probability to detect that there are two latent classes, and the average Rand Index, as a function of sample size (N) and correlItion (r) among indices, over 500 replications. Words in parentheses indicate the R functions used.

performance of the most common clustering methods in detecting the correct number of clusters and classifying human participants using continuous indicators, with an eye on scenarios plausible in cognitive psychology. We found that the median sample size used in published studies (i.e., N = 322) is probably insufficient to detect the correct number of clusters and classify individuals unless in optimal, but unlikely conditions (i.e., in the case of large or very large effect sizes obtained on a large number of informative and preferably orthogonal indicators, i.e., not correlating with each other). This was true regardless of which clustering technique was used. In terms of questionable practices, a problem with the published literature lies in that, in many cases (21%), the number of solutions tested is not clearly reported. Furthermore, in numerous cases (one in three of all studies), the one-cluster solution was not tested, meaning that a single cluster could possibly represent the best solution even in many of the studies

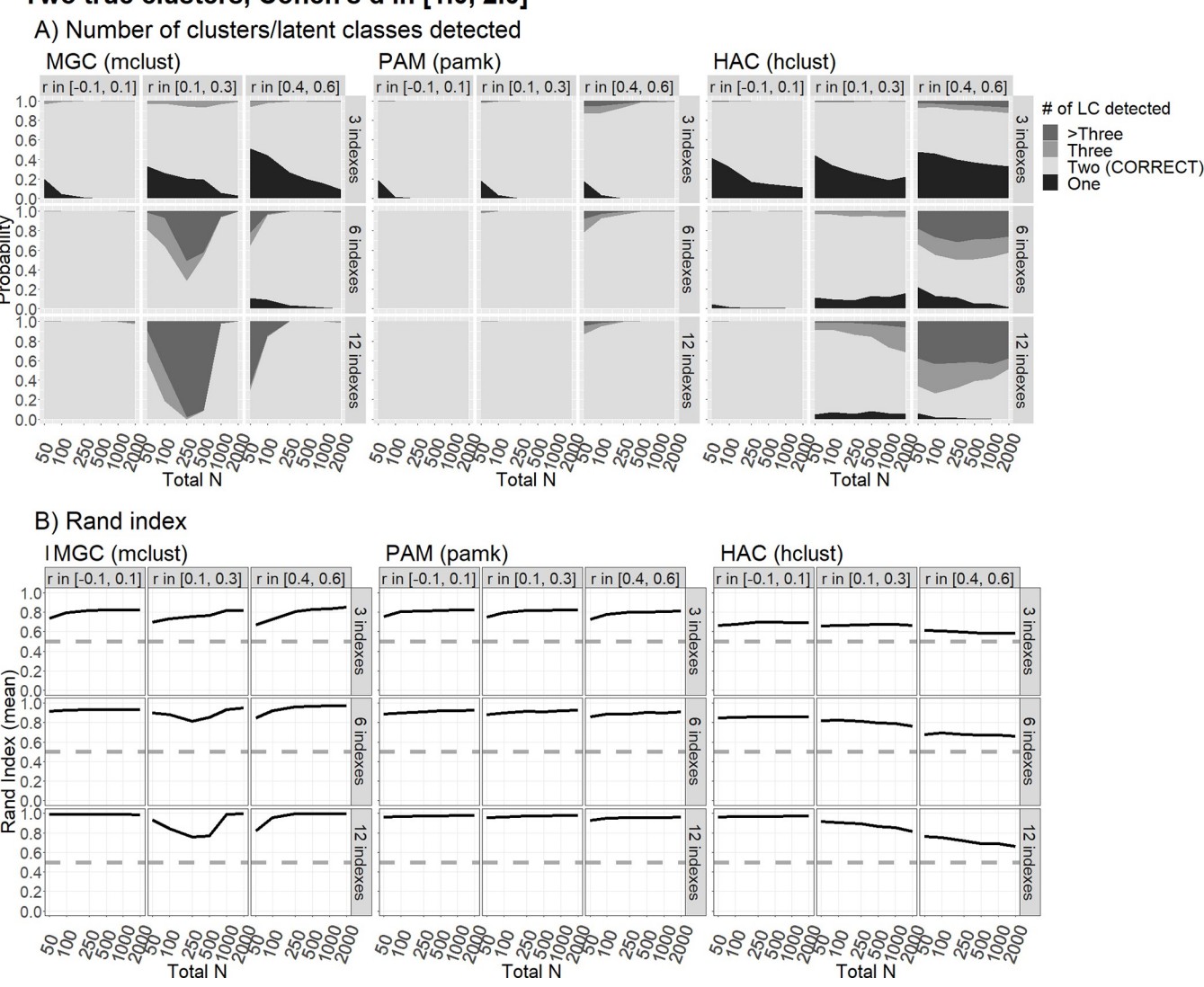

**Fig 5. Clustering performance with two true clusters/latent classes and very large differences.** The figure shows the clustering performance in correctly classifying two true clusters/latent classes differing by a very large Cohen's d [1.0, 2.0] across the indices, using the MGC, PAM and HAC methods, in terms of the probability to detect that there are two latent classes, and the average Rand Index, as a function of sample size (N) and corIelation (r) among indices, over 500 replications. Words in parentheses indicate the R functions used.

reportedly identifying multiple clusters. Somewhat worryingly, all the analyses in the literature reviewed led to the authors concluding for at least two clusters, despite most of them having suboptimal power due to an insufficient sample size under most, if not all plausible conditions (see above). This may be regarded as a case of extreme publication bias. At best, such analyses may be interpreted as having a descriptive purpose, rather than an inferential one. Importantly, dimensionality reduction techniques were almost never used on the entire set of variables before clustering, and in nearly half of all studies the identified clusters tended to differ in terms of "overall low vs overall high" profiles across most or all variables simultaneously, a warning flag that multiple clusters may have been detected due to correlated variables where no sub-populations exist. Finally, very few studies were preregistered, and none of them explicitly performed any power analysis for clustering. Despite that, most studies were published in

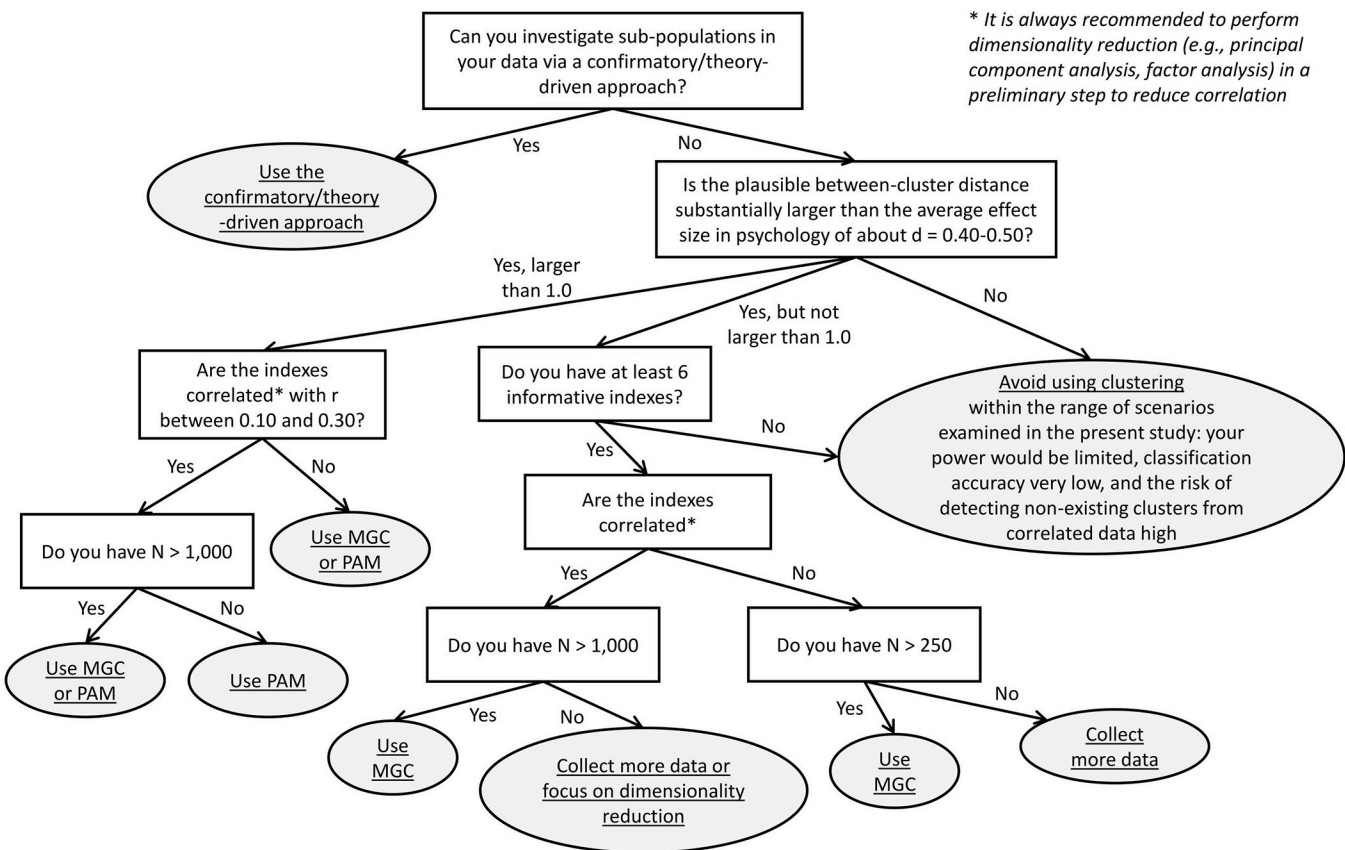

**Fig 6. Decision tree on whether to use clustering based the results of the present study.** The flowchart is limited to the set of scenarios examined in the present study and does not cover all possible alternatives.

high-impact journals (Q1 and Q2 were represented 73% of times considering all studies' categories as indexed by the Clarivate JCR). Most frequently represented categories involved psychiatry, clinical psychology, or developmental or educational psychology.

We then used simulations to perform a design analysis and establish *a priori* the conditions under which clustering techniques could work adequately. Unlike in many articles that we reviewed, we always tested also the single-cluster solution in our simulations, as we considered this as fundamental when performing clustering. Below, we summarize and discuss the results. To further assist the researcher with making decisions based on our results, we propose a (simplified) decision tree in Fig 6. It should be noted that this decision tree is valid only within the set of scenarios examined in the current study.

We found virtually no realistic scenario under which clustering techniques could reliably differentiate between one vs. two real clusters when the effect sizes were small (which, however, might be considered the most plausible scenario for genuine psychological research). This is clear from the largely overlapping plots for "number of clusters detected" in Figs 1 and 2. The only notable exceptions concern the scenarios with 12 informative indicators, but only for large sample sizes (N ≥1,000), and only for MGC. In the latter case, intriguingly, having correlated indicators is not necessarily worse than having orthogonal ones, but only if the correlations are large. In this scenario, the MGC can reveal an adequately informative covariance structure within the clusters. This may represent a crucial advantage of the model-based techniques. With only modest correlations among the indicators, however, there is a higher probability of the algorithm not finding an informative covariance structure, so the procedure may

fail to detect the correct number of clusters. This latter problem is exacerbated with smaller sample sizes (N < 1,000). In fact, regardless of the number of clusters detected, the Rand Index is nearly always at chance level (0.5) when Cohen's $d$s are sampled between 0.20 and 0.60. This would suggest that most of the statistical units are misclassified, even when the right number of (two) clusters is detected.

We recommend that researchers who use clustering methods in cognitive (and more generally psychological) fields always consider the single-cluster solution first before focusing on any multiple-cluster alternative. We believe that bearing the single-cluster solution in mind as a plausible—and perhaps the most plausible—alternative help understand the risk of false positives, meaning the risk of detecting multiple clusters when there is really only one multivariate population. As we have shown, such a risk can be high in many realistic research scenarios. Fig 2 suggests that this risk is substantial and may even be virtually unavoidable in certain conditions—with correlated indicators, for instance. Large correlations (between .40 and .60) are especially problematic when clustering using the PAM or HAC algorithms, and the risk of false positives is paradoxically greater the larger the sample size (though this is simply because, with small sample sizes of 100 or less, the methods lack the power to detect clusters, even if they exist). As for the MGC, the greatest risks emerge for moderate (rather than strong) correlations, and small samples (N < 1,000 for moderate correlations, or N < 250 for large correlations), especially when there are many indicators (6 or 12) involved. Interestingly, unlike the other two methods, the MGC has a tendency to erroneously detect not just two, but even three or more than three clusters when it incurs false positives. Finally, the HAC method generally outperforms the PAM in the one-cluster scenario, especially with moderate correlations. This may be due simply to the HAC having less power to detect multiple clusters than the PAM even when they are there, as shown in the scenarios with Cohen's $d$ indicating large to very large effect sizes.

A consistently good performance for two-cluster scenarios only emerges for very large effect sizes (Cohen's $d$ sampled between 1.0 and 2.0). In this situation, only certain combinations of parameters prove problematic, such as only 3 informative indicators combined with a small sample size (N < 250), or moderate correlations ($\rho$ between .10 and .30) combined with a medium sample size (N between 100 and 500) if the MGC is used. The HAC fails to detect and adequately classify two clusters in some cases, especially when there are only 3 informative indicators, or in the case of large correlations. In all the other cases, a very large Cohen's $d$ leads to an adequate clustering. We argue, however, that this is not what we should set our sights on in modern-day cognitive psychological research, which should contain not only innovative aspects, but also reasonable expectations. This would not be the case, for instance, of two sub-populations with very large differences that go undetected until exploratory, data-driven analyses are performed. We should also carefully consider whether some combinations of parameters are plausible. Is it credible, for instance, to expect many indicators (e.g., 6 or 12) to be simultaneously informative about the clusters and also uncorrelated (e.g. nearly orthogonal)? In other words, could they all contribute to informing about the true underlying clusters, and all do so independently from each other? To give an example, a condition in which two clusters are obtained on 12 indicators, correlated in the range between -.10 and .10, meaning that the two sub-populations simultaneously differ in all 12 nearly-orthogonal measures selected by the researchers—is highly implausible. Such scenarios should be regarded as unrealistic in cognitive psychological research. It might still be plausible, however, for two clusters to differ simultaneously on numerous variables if these variables are moderately or strongly correlated (e.g., because the clusters would simultaneously differ on a set of partly-overlapping cognitive measures). If two clusters differ on several variables, we can probably assume that the variables are at least partially redundant.

A scenario is more plausible when Cohen's *d* comes between 0.6 and 1.0 (a large, but still credible magnitude in the field of genuine research in cognitive psychology). Under such a scenario, clustering performed well in detecting the right number of clusters, and classifying the sample correctly, with a Rand Index approaching .80. This happens only under certain conditions, however, with a sufficient number of informative orthogonal indicators (at least 6) and a sufficient sample size (N < 250 is clearly not advisable). Once again, the MGC outperforms the other two methods, producing results robust to non-orthogonal indicators, and generally better clustering outcomes. The PAM and HAC with the Duda-Hart test may still be able to identify the existence of two clusters correctly in various different conditions, even when the indicators are correlated, but then they fail to classify the statistical units (i.e., the Rand Index nears the chance level). In short, in scenarios with a large (but not very large) Cohen's *d*, the PAM and HAC tend to produce an adequate classification only when there are numerous (12) indicators that are nearly orthogonal, and the sample size is large (N $\geq$ 1,000).

The present study has some limitations that we accepted for ease of interpretation, but they may point to areas worth investigating in future, and readers might want to consider them when performing their own design analysis. For a start, we only simulated data from normal distributions, which may contribute to explaining why the MGC systematically performed better than the other two methods in our simulations. While normality is often an assumption when dealing with psychological data, skewed or non-normal distributions, partial floor/ceiling effects, or the presence of univariate or multivariate outliers may nonetheless be frequent in real research scenarios. To deal with these issues, researchers need to consider model-based clustering with non-Gaussian density [41] or the use of robust clustering procedures [42]. We also assumed, when simulating data from two clusters, that the two underlying sub-populations would be equally numerous, whereas it is more than likely that one sub-population would be somehow smaller than the other(s) in most real cases. With one cluster being very much smaller than the other, however, statistical power would be even smaller than that emerged from our simulations. Additionally, in scenarios with multiple clusters in which one of them is very small, the model selection/optimal clustering solution could find only the most prominent clusters, with minor one(s) being merged with each other or with a close larger cluster. A researcher should a priori evaluate whether this is an acceptable solution. Finally, when we simulated data from two clusters, we took for granted that all the indicators used for clustering would be informative about the class membership. In real-life settings, studies may plausibly include non-informative variables, which may be irrelevant at best, or add noise to the analysis. While all these conditions were chosen for the sake of simplicity, together they may give rise to an ideal scenario that is very unrealistic. In other words, we risk overestimating the performance of the clustering methods considered vis-à-vis their true performance in (suboptimal) real case scenarios. Even so, we found the clustering methods considered largely inadequate for many plausible scenarios in cognitive psychological research.

The three clustering methods we examined are those most often used for empirical research in cognitive psychology, but they differ substantially in some aspects (Sisodia et al., 2012). Partitioning methods are relatively scalable, simple and suitable for datasets with compact spherical clusters that are clearly separate. They suffer from some drawbacks, however, including a high sensitivity to outliers and noise, and a marked overall decline in effectiveness in high-dimensional spaces. HAC methods have an embedded flexibility regarding the level of granularity, and they are well suited to problems involving point linkages, but changing the linkage measure can generate different results. The PAM and HAC methods are both based on a distance matrix that can be prohibitively large for high-dimensional, massive datasets. Density-based clustering methods reveal arbitrarily-shaped clusters of varying sizes, and they are resistant to noise and outliers. Model-based clustering, on the other hand, is sensitive to the shape

of the indicator's distribution, and unsuitable for high-dimensional datasets because of the curse of the dimensionality phenomenon.

It may be that other, possibly novel clustering algorithms could perform better than those considered here. We believe that our study provides some important information for the purpose of prompting the development of new algorithms more appropriate for use in psychological research. For a start, researchers may try using the *parameters* package in R [43], which has a function that simultaneously runs several clustering algorithms, and indicates the most frequently-detected number of clusters. We avoided using this package due to the unsustainable computational cost of applying it to numerous iterations in a simulated design. Beside these considerations, however, we fear that most real-world scenarios in psychological research are simply unsuitable for the successful use of exploratory clustering on individuals. Even if the problem of spurious correlations among indicators is solved, the prevalence of modest true effects (i.e., limited standardized differences between groups) could make it simply impossible to adequately infer the existence of sub-populations reliably from empirical data. It should be noted, however, that standardized mean differences calculated on subgroups classified using cluster analysis may be (unrealistically) large even if no clusters really exist, but this should not mislead the researcher. To give an example, a simple simulation shows that if individuals are classified as being below or above the median of a uniformly-distributed variable $x$, then the standardized mean difference between the two groups on $x$ would be around -3.5 (or -2.6 for a normally-distributed variable)—but interpreting this number would be an extreme case of overestimation of the effect size [e.g., 16]. As in traditional power analyses, researchers should base their expectations on plausible effect sizes, that in psychology rarely exceed a Cohen's $d$ = 1.0, and are usually much smaller [e.g., 13]. In addition, using clustering methods in an effort to identify previously-undetected sub-populations implies that such sub-populations have so far failed to attract attention, which goes to show that any differences between them are hardly likely to be "eye-catching". This means that, unless researchers are approaching a totally new area of investigation, they should probably expect the effects of interest to be small.

## Conclusion

We suggest that unsupervised clustering methods may pose several challenges when seeking previously-undetected sub-populations of individuals within a larger population in most real-world scenarios of psychological research. Generally speaking, if researchers are eager to use such methods for inferential purposes, they should at least demonstrate that they work under realistic conditions by conducting a careful *a priori* design analysis, as done in the case of *a priori* power analysis for traditional hypothesis testing. As regards inferential errors, we have shown that there is a substantial risk not only of failing to detect clusters that do exist, but also of wrongly identifying clusters that do not, especially from correlated multidimensional data, even when the correlation is only modest (i.e., $\rho$ around .20). Since most problems emerge when using many correlated clustering variables, using techniques for dimensionality reduction is strongly recommended [see also 21]. Using clustering for well-motivated descriptive purposes could be less problematic, though it is still important for researchers to consider whether they might be reporting a multiplicity of cases unnecessarily—to return to Ockham's razor.

We suggest that the results of the present study may offer some guidelines for journal editors who must evaluate studies reporting clustering. We recommend that they consider the following: whether the one-cluster solution has been tested; whether the number of alternative solutions tested has been explicitly reported; and whether any technique for dimensionality reduction has been used on the entire set of clustering indexes in a preliminary step.

Additionally, power analysis should be explicitly reported in an article using clustering. We suggest that failing to fulfill this series of points reflect the use of questionable practices in the context of cluster analysis in psychology. Future studies might further investigate how the frequency of such practices in a set of published articles is specifically associated with a higher risk of false positive results, using a simulation approach such as the one that we proposed here. Even after considering all these aspects, however, our set of scenarios presented in Study 2 may offer a useful reference to evaluate the risks associated with using different methods under different assumptions about index correlation and effect sizes. In all cases, it is important to focus on the interpretability of the reported clustering solution, and on the plausibility of the effect sizes (see the Introduction for what values we consider as plausible effect sizes in psychology). Finally, to assist the researchers and editors with decision on whether the use clustering methods is appropriate, we have provided a decision tree in Fig 6 (note that this decision tree is valid only within the set of scenarios explicitly considered in the present study).

As a last note, we argue that researchers in psychology should favor confirmatory approaches with parsimonious theory-driven methods for data analysis whenever possible. In our Introduction, we presented the case of the political polarization in the US to illustrate what we mean. Under certain assumptions, hypothetical researchers who know nothing about such a polarization could "discover" it using clustering methods. In most real case research scenarios, however, any broad differentiation of individuals across discrete or continuous variables could be hypothesized *a priori*, just like the US's political polarization. This being the case, confirmatory methods are generally more powerful than exploratory ones. We found, for example, that clustering methods are practically always poised to fail if Cohen's $d$ is around 0.40 (Fig 3). On the other hand, if the classification of the individuals in a population could be hypothesized in advance, then a difference of $d = 0.40$ would be easy to detect using confirmatory methods, with reasonable sample sizes (e.g., a power of 80% is reached with just 100 individuals per group using the independent-sample t-test, when $d = 0.40$). Alternatively, the relationships among the variables of interest could just be examined along their continua in an overall population treated as unitary (e.g., a power of 80% is reached with a sample of 193 individuals for a single correlation of $r = .20$). Of course, exploratory and confirmatory approaches differ considerably, and address different research questions. Researchers should nonetheless at least consider whether or not they are trying to "let the data speak for themselves" by inappropriately using an exploratory approach as a surrogate for a confirmatory one.

## Supporting information

**S1 Fig. PRISMA flowchart.**
(TIFF)

**S2 Fig. Marginal distributions and pairwise Pearson's correlations among considered variables of the PISA 2015 dataset.** Correlation test: $^*$p$<$0.05, $^{**}$p$<$0.01, $^{***}$p$<$0.001.
(TIFF)

**S3 Fig.** BIC values at the increasing number of latent groups for different structures of the variance-covariance matrix (left) and classification results obtained with 4 groups (right).
(TIFF)

**S4 Fig.** Clusters identified along two principal components (left) and silhouette width plot for the PAM clustering with 2 clusters (right).
(TIFF)

**S5 Fig.** Silhouette width plot for the HAC clustering with 2 clusters (left) and relative dendrogram based on a complete linkage (right). The two selected clusters are highlighted inside the boxes.
(TIFF)

**S1 Table. Clustering performance, one cluster/latent class.** Number of times of correct discover of a unique cluster (K = 1) over 500 replications and Average Rand Index (Mean and SD) by sample size (N), average correlation (r; null = 0, small = 0.2, large = 0.5), number of indicators, and clustering algorithm (Model-based Gaussian Clustering (MGC), Partitioning Around Medoids (PAM), Hierarchical Agglomerative Clustering (HAC)).
(DOCX)

**S2 Table. Clustering performance, two cluster/latent class, small differences.** Number of times of correct discover of a unique cluster (K = 2) over 500 replications with a Cohen d = 0.4, and Average Rand Index (Mean and SD) by sample size (N), average correlation (r; null = 0, small = 0.2, large = 0.5), number of indicators, and clustering algorithm (Model-based Gaussian Clustering (MGC), Partitioning Around Medoids (PAM), Hierarchical Agglomerative Clustering (HAC)).
(DOCX)

**S3 Table. Clustering performance, two cluster/latent class, large differences.** Number of times of correct discover of a unique cluster (K = 2) over 500 replications with a Cohen d = 0.8, and Average Rand Index (Mean and SD) by sample size (N), correlation (r; null = 0, small = 0.2, large = 0.5), number of indicators, and clustering algorithm (Model-based Gaussian Clustering (MGC), Partitioning Around Medoids (PAM), Hierarchical Agglomerative Clustering (HAC)).
(DOCX)

**S4 Table. Clustering performance, two cluster/latent class, very large differences.** Number of times of correct discover of a unique cluster (K = 2) over 500 replications with a Cohen d = 1.5, and Average Rand Index (Mean and SD) by sample size (N), correlation (r; null = 0, small = 0.2,large = 0.5), number of indicators, and clustering algorithm (Model-based Gaussian Clustering (MGC), Partitioning Around Medoids (PAM), Hierarchical Agglomerative Clustering (HAC)).
(DOCX)

## Acknowledgments

We would like to thank the PsicoStat group from the University of Padua for useful advice and support.

## Author Contributions

**Conceptualization:** Enrico Toffalini, Paolo Girardi, Gianmarco Altoè.

**Data curation:** Paolo Girardi, David Giofrè.

**Formal analysis:** Enrico Toffalini, Paolo Girardi, Gianmarco Altoè.

**Methodology:** Enrico Toffalini, Paolo Girardi, David Giofrè.

**Project administration:** Gianmarco Altoè.

**Writing – original draft:** Enrico Toffalini, Paolo Girardi, David Giofrè.

**Writing – review & editing:** David Giofrè, Gianmarco Altoè.

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
