## [Decision Letter · Decision Letter 0]

9 Feb 2022

PONE-D-21-36500Entia Non Sunt Multiplicanda … Shall I Look for Clusters in My Psychological Data? Maybe NotPLOS ONE

Dear Dr. Toffalini,

Thank you for submitting your manuscript to PLOS ONE. After careful consideration, we feel that it has merit but does not fully meet PLOS ONE’s publication criteria as it currently stands. Therefore, we invite you to submit a revised version of the manuscript that addresses the points raised during the review process.

Both reviewers suggested the paper to be revised. Their comments concern several aspects, from literature review to methodlogy to explanation of results.==============================

We look forward to receiving your revised manuscript.

Kind regards,

Maurizio Naldi

Academic Editor

PLOS ONE

Journal Requirements:

2. Thank you for submitting the above manuscript to PLOS ONE. During our internal evaluation of the manuscript, we found significant text overlap between your submission and the following previously published works, some of which you are an author.

- https://uhra.herts.ac.uk/handle/2299/22138

The text that needs to be addressed involves the Results, Discussion, and Conclusion sections.

Please revise the manuscript to rephrase the duplicated text, cite your sources, and provide details as to how the current manuscript advances on previous work. Please note that further consideration is dependent on the submission of a manuscript that addresses these concerns about the overlap in text with published work.

We will carefully review your manuscript upon resubmission, so please ensure that your revision is thorough

Reviewers' comments:

Reviewer's Responses to Questions

**Comments to the Author**

1. Is the manuscript technically sound, and do the data support the conclusions?

Reviewer #1: Yes

Reviewer #2: Partly

2. Has the statistical analysis been performed appropriately and rigorously? 

Reviewer #1: Yes

Reviewer #2: N/A

3. Have the authors made all data underlying the findings in their manuscript fully available?

Reviewer #1: Yes

Reviewer #2: Yes

4. Is the manuscript presented in an intelligible fashion and written in standard English?

Reviewer #1: Yes

Reviewer #2: Yes

5. Review Comments to the Author

Reviewer #1: The authors revealed the potential mistakes that happened in the published psychological reports that used unsupervised methods. The better practices for these methods, according to the authors, should have the conventional estimation of the effect size and test the availability of the minimal dimensions. Their literature review illustrated all the included studies concluded at least 2 dimensions based on an average of 322 individuals. In terms of recommended 80% power, the smallest effect of interest could be 0.3~0.4. The authors conducted the simulated data sets that had one or two exact clusters. Their simulation results on the three clustering methods revealed the potentially high false-positive rates in the circumstance many psychological studies claimed two clusters claimed.

This study clearly addressed the false positive issues under the clustering methods. The public data and codes sufficiently support the other researchers reproduce their results. For the researchers who are using the clustering methods, this study would suggest warnings to avoid inferential mistakes. I have some suggestions for the authors to enhance their arguments.

- General effect size of psychological research: Published meta-analysis has revealed the general effect size for the particular psychological disciplines (Richard et al., 2003 https://doi.org/10.1037/1089-2680.7.4.331;

Schäfer & Schwarz, 2019 https://doi.org/10.3389/fpsyg.2019.00813 ). For example, social psychology has a relatively small effect size compared to the other disciplines ( d = .40~.63). This meta-analytical estimation is close to the small effect size in the authors’ simulations. Authors would consider these studies as supplementary for the settings of the simulations.

- Impacts of clustering methods: Authors did not summarize the sources of the included studies. Although they provide a sheet of all searched papers, readers will wonder where are these studies came from. In which discipline do researchers prefer the clustering methods? Are there some journals encouraging the usage of clustering? How many citations these studies have accumulated in the later published literature? In line with the summary in Table 1, readers will have a picture of which research topics are facing a potentially high false-positive rate.

- Plausible false positive rates: Although simulation results revealed the risk the researchers do not consider the minimal dimensions in use of the clustering methods, readers hardly imagine how much of the possibility the studies concluded at least three clusters are wrong. Authors may assume it is hard to estimate the false-positive rate based on the simulated data and each method has particular assumptions. My suggestion is to compute the prevalence rate of the questionable practices. There are two approaches the authors could consider. One approach is to investigate the questionable practices of the authors of the 191 papers. The other feasible way is to compute the ratio of the error frequency in simulated data against the number of multiple clusters in published outcomes.

- Recommend practices for researchers and journal editors: The latest conclusion has emphasized the better practices of clustering methods for psychological researchers. In addition to the researchers who are using the cluster methods, journal editors should have the awareness because they suggest the authors conduct dimension reduction checks at the early time of the publication process. With my suggestion about the impacts of clustering methods, the authors could provide special medicine to overcome the publication bias.

I found one minor error as below.

- p. 15 In the setting of pairwise correlations, this article said “(-0.10, 0.10), (0.20, 0.40) and (0.40, 0.60)”. It should be associated with the correlations 0, 0.2, 0.5. However, I believed “(0.20, 0.40)” should be “(0.10, 0.30)”. There is evidence that Fig. 2 ~ Fig. 5 showed “(0.10, 0.30)” instead of “(0.20, 0.40)”.

Reviewer #2: This paper is well written and concerns an important topic in the social sciences, where cluster approaches to identifying groups that differ on psychological constructs such as attitudes and beliefs are gaining popularity. Model selection is crucial in any clustering approach, and knowing the limitations of finding the appropriate number of clusters in different scenarios typically encountered in psychological research is of great importance. My main criticism is that I think the authors assume too much prior knowledge on the part of the readers and that they are not clear enough about what kind of data and clustering approaches they are focusing on. Other concerns relate to the message the authors convey in the general discussion and in the title. Below I explain my concerns in more detail and provide suggestions.

Major comments:

1. Considering that the paper is addressed to psychological researchers, I think that some concepts need further explanation.

a. It is not immediately clear what exactly the authors mean by "psychological data". The authors clarify quite late that they focus on cognitive and neuropsychological studies. In my opinion, this should already be clear from the title, abstract and introduction (knowing exactly what kind of data the authors focus on is also important when talking about "publication bias"; otherwise it gives a wrong picture).

b. I think it is asking too much to assume that readers are familiar with the similarities and differences of the methods under investigation, so it would be important to provide a non-technical intuition for differences and similarities. There was some information in the discussion, but I would have liked to see it already in the introduction of the methods (and then with examples to help understand when the three methods might be used in cognitive and neuropsychological studies; because, as far as I understand, the different methods are useful in different contexts). To address both points, I suggest specifying the nature of the psychological data early on and providing three examples (e.g., from the studies in the literature review) in which the three types of clustering approaches were used. As a side effect, I think it would also make it clearer what exactly is meant by orthogonal items and why this may be unrealistic in many studies (which is currently quite vague, especially if you have only read the introduction).

2. The title and discussion give the impression that the authors advise against clustering approaches for psychological data. In my opinion, any advice should only be given for the specific type of psychological data and cluster approaches that were examined in the literature review and simulation study. I also believe that the message should be a little more nuanced. Of course, it is important to make researchers aware of the current limitations in model selection under certain (realistic) conditions and for certain clustering approaches, but to abandon the idea of clustering in all psychological data is rather strict. Instead, I think the authors should also emphasize that methodologists should take up the challenge and develop better tools for model selection.

3. I personally think that something like a decision tree for researchers or a slightly more concise presentation of the main findings would be very helpful.

Minor comments:

1. The authors note that a major problem is the plausible effect size, which tends to be small in psychological studies. However, from the text, it is not entirely clear what the stated values mean for the type of data/cluster analysis considered in this paper. In this context, "unavoidable measurement error" and "suspicious/trivial looking studies" are briefly mentioned in the same paragraph, which is rather abstract and should also be put in the context of cluster analyses in psychological data.

2. The authors briefly mention the limitation that equal cluster sizes were used in their simulation study. However, it would be nice to have one or two additional sentences describing possible implications (just one possible scenario that I thought of: If there are three real clusters and one is very small, the model selection could find only the most prominent clusters, so that the observations of the small cluster would be merged with the other two clusters).

3. The information about the technical details (computational costs, etc.) is not well placed in the discussion. I suggest moving them to the simulation study section.

4. Throughout the paper I miss the discussion/suggestion to consider interpretability in the final model selection. For example, comparing a one-cluster solution with a two-cluster solution could show whether the differences found are theoretically/empirically meaningful or not. Such a proposal would also raise awareness that data-driven cluster approaches (or model selection) should not be blindly trusted. In their discussion, the authors made the recommendation to pay particular attention to false positives. I think this recommendation and the recommendation to also investigate single-cluster solutions (and to consider interpretability in general) can easily be combined.

6. PLOS authors have the option to publish the peer review history of their article (what does this mean?). If published, this will include your full peer review and any attached files.

Reviewer #1: **Yes: **Sau-Chin Chen

Reviewer #2: No

---

## [Author Response · Author response to Decision Letter 0]

23 Feb 2022

EDITOR

RESPONSE: We have now carefully re-formatted our manuscript to follow PLOS ONE’s style requirements.

2. Thank you for submitting the above manuscript to PLOS ONE. During our internal evaluation of the manuscript, we found significant text overlap between your submission and the following previously published works, some of which you are an author. - https://uhra.herts.ac.uk/handle/2299/22138 The text that needs to be addressed involves the Results, Discussion, and Conclusion sections. We would like to make you aware that copying extracts from previous publications, especially outside the methods section, word-for-word is unacceptable. In addition, the reproduction of text from published reports has implications for the copyright that may apply to the publications. Please revise the manuscript to rephrase the duplicated text, cite your sources, and provide details as to how the current manuscript advances on previous work. Please note that further consideration is dependent on the submission of a manuscript that addresses these concerns about the overlap in text with published work. We will carefully review your manuscript upon resubmission, so please ensure that your revision is thorough

RESPONSE: We have looked closely at the linked paper (https://uhra.herts.ac.uk/handle/2299/22138). In fact, none of us is the author of that PhD Thesis nor had ever heard of it before. The thesis is on a topic very far away from that of the current report, although it uses clustering methods. We could not find where the text overlap was (we assume it may regard the description of a clustering method). After the current review, we hope that no text overlap will emerge again. However, in such a case, could you please specifically point us to which sections of the text overlap?

REVIEWER #1

The authors revealed the potential mistakes that happened in the published psychological reports that used unsupervised methods. The better practices for these methods, according to the authors, should have the conventional estimation of the effect size and test the availability of the minimal dimensions. Their literature review illustrated all the included studies concluded at least 2 dimensions based on an average of 322 individuals. In terms of recommended 80% power, the smallest effect of interest could be 0.3~0.4. The authors conducted the simulated data sets that had one or two exact clusters. Their simulation results on the three clustering methods revealed the potentially high false-positive rates in the circumstance many psychological studies claimed two clusters claimed.

This study clearly addressed the false positive issues under the clustering methods. The public data and codes sufficiently support the other researchers reproduce their results. For the researchers who are using the clustering methods, this study would suggest warnings to avoid inferential mistakes. I have some suggestions for the authors to enhance their arguments.

RESPONSE: We thank the reviewer for this good summary of our work.

- General effect size of psychological research: Published meta-analysis has revealed the general effect size for the particular psychological disciplines (Richard et al., 2003 https://doi.org/10.1037/1089-2680.7.4.331;

Schäfer & Schwarz, 2019 https://doi.org/10.3389/fpsyg.2019.00813 ). For example, social psychology has a relatively small effect size compared to the other disciplines (d = .40~.63). This meta-analytical estimation is close to the small effect size in the authors’ simulations. Authors would consider these studies as supplementary for the settings of the simulations.

RESPONSE: We thank the reviewer for this suggestion. We have now cited these two studies to strengthen our argument about the chosen effect sizes.

- Impacts of clustering methods: Authors did not summarize the sources of the included studies. Although they provide a sheet of all searched papers, readers will wonder where are these studies came from. In which discipline do researchers prefer the clustering methods? Are there some journals encouraging the usage of clustering? How many citations these studies have accumulated in the later published literature? In line with the summary in Table 1, readers will have a picture of which research topics are facing a potentially high false-positive rate.

RESPONSE: We thank the reviewer for this suggestion. We have now added a paragraph in the Results (and a comment in the General Discussion), about the characteristics of the journals in which the articles were published. We found that the median impact factor was 3.23, and the journals tended to have a high impact within their JCR categories (Q1 and Q2 were represented 73% of times). We avoided counting the citations accumulated by single articles as they are strongly time-dependent (e.g., whether the article was published in 2016 or 2020). Additionally, we found that the most represented categories were Clinical psychology/psychiatry, and Developmental and Educational psychology. Full data reporting this information has been uploaded on OSF.

- Plausible false positive rates: Although simulation results revealed the risk the researchers do not consider the minimal dimensions in use of the clustering methods, readers hardly imagine how much of the possibility the studies concluded at least three clusters are wrong. Authors may assume it is hard to estimate the false-positive rate based on the simulated data and each method has particular assumptions. My suggestion is to compute the prevalence rate of the questionable practices. There are two approaches the authors could consider. One approach is to investigate the questionable practices of the authors of the 191 papers. The other feasible way is to compute the ratio of the error frequency in simulated data against the number of multiple clusters in published outcomes.

RESPONSE: If we understand correctly, the reviewer suggests that we formally examine questionable practices in the context of the use of cluster analysis. We have now explicitly listed what we considered as questionable practices in this context (see [Sec sec002], Method, Coding of the Studies), and described the results accordingly. Based on the results, cluster analysis questionable practices are more of a rule than an exception in the literature. As for the alternative way that you propose, we believe that computing false positive rates from the reviewed studies is not possible, because in the large majority of cases we lack a crucial piece of information, namely the set of intercorrelations among indexes. For this reason, we suggest that our results should be taken just as a reference to evaluate the risk of a single study under possible alternative scenarios, rather than a tool to compute false positive rates in the literature (a few lines have been added also in the Conclusions, in the paragraph offering recommendation for journal editors as suggested in the point below).

- Recommend practices for researchers and journal editors: The latest conclusion has emphasized the better practices of clustering methods for psychological researchers. In addition to the researchers who are using the cluster methods, journal editors should have the awareness because they suggest the authors conduct dimension reduction checks at the early time of the publication process. With my suggestion about the impacts of clustering methods, the authors could provide special medicine to overcome the publication bias.

RESPONSE: We have added a new paragraph in the Conclusions, to offer recommendations for journal editors who want to evaluate whether the use of clustering was appropriate in a specific study.

I found one minor error as below.

- p. 15 In the setting of pairwise correlations, this article said “(-0.10, 0.10), (0.20, 0.40) and (0.40, 0.60)”. It should be associated with the correlations 0, 0.2, 0.5. However, I believed “(0.20, 0.40)” should be “(0.10, 0.30)”. There is evidence that Fig. 2 ~ Fig. 5 showed “(0.10, 0.30)” instead of “(0.20, 0.40)”.

RESPONSE: We thank the reviewer for his careful reading. We have amended this point.

REVIEWER #2

This paper is well written and concerns an important topic in the social sciences, where cluster approaches to identifying groups that differ on psychological constructs such as attitudes and beliefs are gaining popularity. Model selection is crucial in any clustering approach, and knowing the limitations of finding the appropriate number of clusters in different scenarios typically encountered in psychological research is of great importance. My main criticism is that I think the authors assume too much prior knowledge on the part of the readers and that they are not clear enough about what kind of data and clustering approaches they are focusing on. Other concerns relate to the message the authors convey in the general discussion and in the title. Below I explain my concerns in more detail and provide suggestions.

RESPONSE: We thank the reviewer for the positive consideration of our work.

Major comments:

1. Considering that the paper is addressed to psychological researchers, I think that some concepts need further explanation.

a. It is not immediately clear what exactly the authors mean by "psychological data". The authors clarify quite late that they focus on cognitive and neuropsychological studies. In my opinion, this should already be clear from the title, abstract and introduction (knowing exactly what kind of data the authors focus on is also important when talking about "publication bias"; otherwise it gives a wrong picture).

RESPONSE: We have now clarified that we focused on the cognitive literature. This has been amended in the title, abstract, and introduction.

b. I think it is asking too much to assume that readers are familiar with the similarities and differences of the methods under investigation, so it would be important to provide a non-technical intuition for differences and similarities. There was some information in the discussion, but I would have liked to see it already in the introduction of the methods (and then with examples to help understand when the three methods might be used in cognitive and neuropsychological studies; because, as far as I understand, the different methods are useful in different contexts). To address both points, I suggest specifying the nature of the psychological data early on and providing three examples (e.g., from the studies in the literature review) in which the three types of clustering approaches were used. As a side effect, I think it would also make it clearer what exactly is meant by orthogonal items and why this may be unrealistic in many studies (which is currently quite vague, especially if you have only read the introduction).

RESPONSE: We agree with the reviewer, an example can provide a clarification of the methods that we have discussed. In the Methods section (see Clustering Procedures), we have offered an example of an application starting from a free available dataset in R (most papers considered did not add a data repository), explaining how the three algorithms work. Additionally, we have briefly commented their output at the end of the Results section, with reference to the results of our simulations. The code for this example has been also uploaded on OSF. Additionally, we have specified that the meaning of “orthogonal” indicators implies and assumption about the absence of any correlation among them after controlling for possible cluster membership.

2. The title and discussion give the impression that the authors advise against clustering approaches for psychological data. In my opinion, any advice should only be given for the specific type of psychological data and cluster approaches that were examined in the literature review and simulation study. I also believe that the message should be a little more nuanced. Of course, it is important to make researchers aware of the current limitations in model selection under certain (realistic) conditions and for certain clustering approaches, but to abandon the idea of clustering in all psychological data is rather strict. Instead, I think the authors should also emphasize that methodologists should take up the challenge and develop better tools for model selection.

RESPONSE: We have now offered a more nuanced message throughout the manuscript (and we have even edited the title accordingly). Also, we have added a paragraph in the Conclusions to offer specific recommendations to editors and researchers who want to evaluate whether using clustering is appropriate in an article.

3. I personally think that something like a decision tree for researchers or a slightly more concise presentation of the main findings would be very helpful.

RESPONSE: We thank the reviewer for this suggestion. We have now added a decision tree (Fig 6) to assist the researcher with making decisions on whether and how to use clustering.

Minor comments:

1. The authors note that a major problem is the plausible effect size, which tends to be small in psychological studies. However, from the text, it is not entirely clear what the stated values mean for the type of data/cluster analysis considered in this paper. In this context, "unavoidable measurement error" and "suspicious/trivial looking studies" are briefly mentioned in the same paragraph, which is rather abstract and should also be put in the context of cluster analyses in psychological data.

RESPONSE: We have now clarified what we meant by plausible effect size, also adding further references. We have also specified that these are only educated guess based on the general literature, however, and that they may vary when considering specific research sub-fields. In addition, we have offered a decision tree (Fig 6) with explicit benchmarks for the effect sizes values.

2. The authors briefly mention the limitation that equal cluster sizes were used in their simulation study. However, it would be nice to have one or two additional sentences describing possible implications (just one possible scenario that I thought of: If there are three real clusters and one is very small, the model selection could find only the most prominent clusters, so that the observations of the small cluster would be merged with the other two clusters).

RESPONSE: We have now added a few sentences to further clarify alternative scenarios with unbalanced clusters or several clusters, as suggested by the reviewer.

3. The information about the technical details (computational costs, etc.) is not well placed in the discussion. I suggest moving them to the simulation study section.

RESPONSE: The information has been relocated as suggested.

4. Throughout the paper I miss the discussion/suggestion to consider interpretability in the final model selection. For example, comparing a one-cluster solution with a two-cluster solution could show whether the differences found are theoretically/empirically meaningful or not. Such a proposal would also raise awareness that data-driven cluster approaches (or model selection) should not be blindly trusted. In their discussion, the authors made the recommendation to pay particular attention to false positives. I think this recommendation and the recommendation to also investigate single-cluster solutions (and to consider interpretability in general) can easily be combined.

RESPONSE: We have further articulated these recommendations as suggested. A note on the importance of the interpretability of a cluster solution has also been added in the Conclusions. Also, we have further clarified that we always considered the single-cluster solution in our simulations, and that even when the single-cluster solution is explicitly considered, there is still a high risk of false positives in many possible scenarios.

---

## [Decision Letter · Decision Letter 1]

25 Apr 2022

PONE-D-21-36500R1Entia Non Sunt Multiplicanda … Shall I Look for Clusters in My Cognitive Data?PLOS ONE

Dear Dr. Toffalini,

Thank you for submitting your manuscript to PLOS ONE. After careful consideration, we feel that it has merit but does not fully meet PLOS ONE’s publication criteria as it currently stands. Therefore, we invite you to submit a revised version of the manuscript that addresses the points raised during the review process.

One of the reviewers has provided two suggestions to improve the paper. Please try to cope with them.==============================

We look forward to receiving your revised manuscript.

Kind regards,

Maurizio Naldi

Academic Editor

PLOS ONE

Journal Requirements:

Reviewers' comments:

Reviewer's Responses to Questions

**Comments to the Author**

1. If the authors have adequately addressed your comments raised in a previous round of review and you feel that this manuscript is now acceptable for publication, you may indicate that here to bypass the “Comments to the Author” section, enter your conflict of interest statement in the “Confidential to Editor” section, and submit your "Accept" recommendation.

Reviewer #1: All comments have been addressed

Reviewer #2: All comments have been addressed

2. Is the manuscript technically sound, and do the data support the conclusions?

Reviewer #1: Yes

Reviewer #2: Yes

3. Has the statistical analysis been performed appropriately and rigorously? 

Reviewer #1: Yes

Reviewer #2: Yes

4. Have the authors made all data underlying the findings in their manuscript fully available?

Reviewer #1: Yes

Reviewer #2: Yes

5. Is the manuscript presented in an intelligible fashion and written in standard English?

Reviewer #1: Yes

Reviewer #2: Yes

6. Review Comments to the Author

Reviewer #1: The authors had solved the primary issues in my previous comments. There are two minor issues the editor would decide if the authors have to answer them.

1. I appreciate the authors presented more summary about their collected papers. On the other hand, the latest title is "… Shall I look for clusters in my cognitive data?". Obviously no cognitive papers were collected in the authors' review. Authors would consider the key terms in the title or emphasize the constraints in this article.

2. I agree the authors' clarifications. Indeed we require the published data to estimate the false positive rate (FA). We have known high FA is the primary consequence of QRPs. I am curious if this simulation framework could be useful to estimate a latent false positive rate when the questionable practices are frequently identified in a set of papers. I understand this work is beyond the scope of this study. Still I encourage the authors could describe the potential every researcher of clustering methods use this simulation framework to monitor their practices in data analysis.

Reviewer #2: All comments have been addressed satisfactorily. I specifically appreciate the decision tree and the explanation of the methods using an existing dataset (thank you for providing the R code).

One final note: "In the present study, we examined the conditions under which popular techniques for clustering individuals may plausibly work on cognitive data, with a special focus on cognitive research."  "with a special focus on cognitive research" is redundant now.

7. PLOS authors have the option to publish the peer review history of their article (what does this mean?). If published, this will include your full peer review and any attached files.

Reviewer #1: **Yes: **Sau-Chin Chen

Reviewer #2: No

---

## [Author Response · Author response to Decision Letter 1]

3 May 2022

Reviewer #1: The authors had solved the primary issues in my previous comments. There are two minor issues the editor would decide if the authors have to answer them.

RESPONSE: We thank the reviewer for his positive comment. His suggestions in the previous round of revision were very useful and contributed to improve the manuscript.

1. I appreciate the authors presented more summary about their collected papers. On the other hand, the latest title is "… Shall I look for clusters in my cognitive data?". Obviously no cognitive papers were collected in the authors' review. Authors would consider the key terms in the title or emphasize the constraints in this article.

RESPONSE: We prefer keeping the title as it is because it is the result of the previous round of revision and a specific suggestion from Reviewer 2. Indeed, as emphasized throughout the paper, our aims, analyses, and results are focused on quantitative cognitive psychology data. However, we are open to edit it and remove “in my cognitive data” or change it with “in my quantitative psychological data” if the Editor beliefs that this is really the best choice.

2. I agree the authors' clarifications. Indeed we require the published data to estimate the false positive rate (FA). We have known high FA is the primary consequence of QRPs. I am curious if this simulation framework could be useful to estimate a latent false positive rate when the questionable practices are frequently identified in a set of papers. I understand this work is beyond the scope of this study. Still I encourage the authors could describe the potential every researcher of clustering methods use this simulation framework to monitor their practices in data analysis.

RESPONSE: This point is very interesting, and it has the potential to open further venue of research. We have added a comment on how future studies using a simulation approach could help elucidate the impact of questionable research practices on the results obtained from clustering methods (see Discussion).

Reviewer #2: All comments have been addressed satisfactorily. I specifically appreciate the decision tree and the explanation of the methods using an existing dataset (thank you for providing the R code).

RESPONSE: We thank the reviewer for this positive comment.

One final note: "In the present study, we examined the conditions under which popular techniques for clustering individuals may plausibly work on cognitive data, with a special focus on cognitive research."  "with a special focus on cognitive research" is redundant now.

RESPONSE: This sentence has now been corrected. Thank you very much for the careful reading of our manuscript.

---

## [Decision Letter · Decision Letter 2]

25 May 2022

Entia Non Sunt Multiplicanda … Shall I Look for Clusters in My Cognitive Data?

PONE-D-21-36500R2

Dear Dr. Toffalini,

We’re pleased to inform you that your manuscript has been judged scientifically suitable for publication and will be formally accepted for publication once it meets all outstanding technical requirements.

Kind regards,

Maurizio Naldi

Academic Editor

PLOS ONE

Additional Editor Comments (optional):

Reviewers' comments:

Reviewer's Responses to Questions

**Comments to the Author**

1. If the authors have adequately addressed your comments raised in a previous round of review and you feel that this manuscript is now acceptable for publication, you may indicate that here to bypass the “Comments to the Author” section, enter your conflict of interest statement in the “Confidential to Editor” section, and submit your "Accept" recommendation.

Reviewer #1: All comments have been addressed

Reviewer #2: All comments have been addressed

2. Is the manuscript technically sound, and do the data support the conclusions?

Reviewer #1: Yes

Reviewer #2: Yes

3. Has the statistical analysis been performed appropriately and rigorously? 

Reviewer #1: Yes

Reviewer #2: Yes

4. Have the authors made all data underlying the findings in their manuscript fully available?

Reviewer #1: Yes

Reviewer #2: Yes

5. Is the manuscript presented in an intelligible fashion and written in standard English?

Reviewer #1: Yes

Reviewer #2: Yes

6. Review Comments to the Author

Reviewer #1: I accepted the authors' responses to my comments. I have no more comments for the latest revision.

Reviewer #2: (No Response)

7. PLOS authors have the option to publish the peer review history of their article (what does this mean?). If published, this will include your full peer review and any attached files.

Reviewer #1: **Yes: **Sauchin Chen

Reviewer #2: No

---

## [Editor Report · Acceptance letter]

21 Jun 2022

PONE-D-21-36500R2 

*Entia Non Sunt Multiplicanda* … Shall I look for clusters in my cognitive data? 

Dear Dr. Toffalini:

I'm pleased to inform you that your manuscript has been deemed suitable for publication in PLOS ONE. Congratulations! Your manuscript is now with our production department. 

Kind regards, 

on behalf of

Professor Maurizio Naldi 

Academic Editor

PLOS ONE